# High-efficiency green management of potato late blight by a self-assembled multicomponent nano-bioprotectant

Yuxi Wang[1,6], Mingshan Li[1,6], Jiahan Ying[1], Jie Shen [1], Daolong Dou [1,2], Meizhen Yin [3], Stephen C. Whisson[4], Paul R. J. Birch[4,5], Shuo Yan[1] ✉ & Xiaodan Wang [1] ✉

Potato late blight caused by *Phytophthora infestans* is a devastating disease worldwide. Unlike other plant pathogens, double-stranded RNA (dsRNA) is poorly taken up by *P. infestans*, which is a key obstacle in using dsRNA for disease control. Here, a self-assembled multicomponent nano-bioprotectant for potato late blight management is designed based on dsRNA and a plant elicitor. Nanotechnology overcomes the dsRNA delivery bottleneck for *P. infestans* and extends the RNAi protective window. The protective effect of nano-enabled dsRNA against infection arises from a synergistic mechanism that bolsters the stability of dsRNA and optimizes its effective intracellular delivery. Additionally, the nano-enabled elicitor enhances endocytosis and amplifies the systemic defense response of the plants. Co-delivery of dsRNA and an elicitor provides a protective effect via the two aspects of pathogen inhibition and elevated plant defense mechanisms. The multicomponent nano-bioprotectant exhibits superior control efficacy compared to a commercial synthetic pesticide in field conditions. This work proposes an eco-friendly strategy to manage devastating plant diseases and pests.

Late blight, caused by the oomycete *Phytophthora infestans*, is the most economically important potato disease worldwide. It triggered the Irish Famine in the mid-19th century and still causes economic losses in excess of $6 billion annually[1,2]. Taking China as an example, according to the official statistics from the Ministry of Agriculture and Rural Affairs of the People's Republic of China, the occurrence of potato late blight affects up to 50.91% of the total cultivated area, and the annual yield loss accounts for between 12.56 and 583.10 kilotons. A large outbreak of late blight occurred in 2012, and the total infected area peaked at 2.65 million ha. Potato production relies almost exclusively on chemical measures to control late blight. However, the frequent application of synthetic pesticides has raised environmental,

health and safety concerns[3,4]. Therefore, alternative green technologies should be developed for reduced toxicity, more target specificity and environmental sustainability. RNA interference (RNAi) is a conserved regulatory mechanism mediated by double-stranded RNA (dsRNA), which can silence or inhibit the expression of target genes[5–8]. Crops can be directly sprayed with dsRNA (spray-induced gene silencing, SIGS) targeting key genes of plant pathogens to induce specific silencing, thus providing an opportunity to realize sustainable eco-friendly disease management[9–12].

Although fungi such as *Botrytis cinerea* can efficiently absorb dsRNA from the environment[13], *P. infestans* does not readily take up environmental dsRNA[14,15], possibly due to characteristics of its cell

[1]College of Plant Protection, China Agricultural University, Beijing 100193, China. [2]College of Plant Protection, Academy for Advanced Interdisciplinary Studies, Nanjing Agricultural University, Nanjing 210095, China. [3]State Key Laboratory of Chemical Resource Engineering, Beijing Lab of Biomedical Materials, Beijing University of Chemical Technology, Beijing 100029, China. [4]Cell and Molecular Sciences, James Hutton Institute, Errol Road, Invergowrie, Dundee DD2 5DA, UK. [5]Division of Plant Sciences, School of Life Science, University of Dundee (at James Hutton Institute), Invergowrie, Dundee DD2 5DA, UK. [6]These authors contributed equally: Yuxi Wang, Mingshan Li. ✉e-mail: yanshuo2011@foxmail.com; xdwang@cau.edu.cn

wall[16,17]. In oomycetes, low uptake efficiency of exogenous RNA is the key obstacle in identifying gene function and RNA fungicide application[13,18,19]. Thus, a SIGS strategy requires a reliable vehicle to break the dsRNA delivery bottleneck for *P. infestans*. Nanoparticles with good biocompatibility can promote the translocation of exogenous substances, and have been widely adopted in cancer therapy[20–23]. Our previous study demonstrated that nanocarriers can accelerate endocytosis to promote the translocation of dsRNA across cell membranes for efficient cellular uptake[24–26].

We previously constructed a facile-synthesized star polycation (SPc) as a low cost dsRNA nanocarrier, with high intracellular delivery efficiency, good biocompatibility and biodegradability[27,28]. The SPc nanocarrier consists of a hydrophobic core and a hydrophilic shell with positively-charged tertiary amine in the side chain. The hydrophobic core can be applied to assemble hydrophobic active ingredients, and the tertiary amine can combine with negatively-charged nucleic acids through electrostatic adsorption[29].

In this study, we find that the SPc overcomes the dsRNA delivery bottleneck for *P. infestans*, thus efficiently inhibiting infection and extending the RNAi protective window. The SPc nanocarrier enhances the stability of dsRNA, protecting it from degradation, and facilitates its efficient intracellular delivery. Moreover, we co-express dsRNAs of two target genes in a bacterial system to markedly increase production at low cost to meet the demands necessary for field application. In addition, the nano-enabled elicitor amplifies the systemic defense responses of plants. After a simple incubation process, SPc can self-assemble with dsRNA and elicitors into bioprotectant nanoparticles through hydrophobic and electrostatic forces. The multicomponent nano-bioprotectant provides a significant protective effect in both laboratory and field conditions, enabling sustainable green management of potato late blight.

## Results and discussion

### Protective effect of SPc on dsRNA and its enhanced delivery
In this study, we demonstrated that SPc could protect dsRNA from enzymolysis whereas naked dsRNA was degraded quickly by RNase A at low salt concentration[30], thus increasing the environmental stability of dsRNA (Supplementary Fig. 1). To determine uptake by *P. infestans*, naked ds*eGFP* (dsRNA targeting the *enhanced green fluorescent protein* gene) was labeled with fluorescein to yield green fluorescence. After assembly with SPc, the fluorescence intensity of ds*eGFP* was essentially unchanged (Supplementary Fig. 2). Confocal microscopy observations showed that naked ds*eGFP* was poorly absorbed by *P. infestans* sporangia and hyphae. In comparison, SPc facilitated the uptake of ds*eGFP*, overcoming the dsRNA delivery bottleneck (Fig. 1a). When applied to plant surfaces, the fluorescence intensity of SPc-loaded ds*eGFP* was increased by 3.65-fold compared to naked ds*eGFP* in potato leaves, which was consistently observed in both roots and leaves of the model solanaceous host plant *Nicotiana benthamiana* (Fig. 1b–d). Similarly, sporangia of a transgenic *P. infestans* strain expressing tdTomato (tandem dimer Tomato) fluorescent protein were treated with ds*tdTomato*/SPc complex and ds*tdTomato* alone. The red fluorescence was markedly reduced in the ds*tdTomato*/SPc complex treatment, and the *tdTomato* gene expression level was decreased by 2.7-fold compared to ds*tdTomato* alone (Fig. 1e, f). These results supported that the dsRNA could be effectively delivered into both oomycetes and plants with the help of SPc, leading to higher RNAi efficiency. Since SPc can protect dsRNA from enzymatic degradation and promote dsRNA delivery, the elevated uptake might be due to the combination of these two aspects.

### Nano-enabled dsRNA exhibits a significant protective effect
Oomycetes such as *P. infestans* employ a complex assortment of cell wall-degrading enzymes to breach plant cells, with the cuticle being the first barrier. Once infection has been successfully established, hyphae subsequently grow through the plant tissue to extend haustoria into cells, thus potentially facilitating uptake of nutrients from plant cells[16]. During the initial stages of the infection process, cutinase (*PiCut3*) degrades cutin as the outermost pathogen-plant barrier[31]. The haustorial membrane protein 1 (PiHmp1) is a jacalin lectin-like protein that localizes to the haustorium membrane and is essential for infection[32]. We synthesized dsRNAs for *PiHmp1* (216 bp) and *PiCut3* (251 bp) in vitro, and then sprayed ds*PiHmp1*, ds*PiCut3* and their SPc-loaded complexes onto detached potato leaves. *P. infestans* sporangia suspension was inoculated onto plant leaves 24 h later, and the lesion area was measured at 2, 5, and 15 days post infection (dpi). As shown in Fig. 1g, h, application of SPc-loaded ds*PiHmp1* or ds*PiCut3* strongly protected the leaves against infection after 24 h, whereas naked dsRNA had no protective effect due to limited uptake and instability of dsRNA, which was consistent with the report of Qiao et al.[15]. Other studies using naked dsRNAs showed decreased infection by *P. infestans*, but not complete protection against disease[14]. Importantly, there were still no disease symptoms in the leaves after 15 days of initial inoculation, prolonging the RNAi protective window. Our study found that the SPc/dsRNA complex remained on plants for up to 12 days and was 1.5-fold more abundant than dsRNA alone (Supplementary Fig. 3). We synthesized dsRNA from the two genes together in tandem (ds*PiHmp1+PiCut3;* 24 ng/μL) without SPc loaded in vitro, and showed that this had a significant silencing effect on both *PiHmp1* and *PiCut3* (Supplementary Fig. 4). These results demonstrated that dsRNAs homologous to two selected target genes could be combined and applied to inhibit *P. infestans* infection, but only via the SPc-based delivery system.

Scaling up production of dsRNA is a limiting factor for field applications; thus we established a bacterial expression system based on pET28-BL21(DE3) RNase III-. Its dsRNA expression is three-fold greater than that of the widely-used dsRNA expression system L4440-HT115 (DE3), which has increased utility for large-scale dsRNA production with low cost[33]. As shown in Supplementary Fig. 5a, b, ds*PiHmp1* and ds*PiCut3* were tandemly constructed in the pET28a vector and co-expressed in our bacterial expression system. The bacteria were lysed using lysozyme, incubated with SPc, and then analyzed for protective effects on leaves. The RNAi efficiency of SPc-loaded ds*PiHmp1+PiCut3* produced by the bacterial system was comparable to in vitro-synthesized dsRNA, and the detached potato leaves treated with ds*PiHmp1+PiCut3*/SPc complex showed no disease symptoms, confirming its protective effect (Supplementary Fig. 5c–e).

### Nano-enabled elicitor amplifies the defense responses of plants
To further increase the control efficacy of the ds*PiHmp1+PiCut3*/SPc complex for field application, a plant elicitor cellobiose (4-OR-β-D-glucopyranosyl-β-D-glucopyranose) was introduced to construct a multicomponent nano-bioprotectant. Studies have shown that cellobiose can be effectively sensed by plants to activate defense signaling against diseases[34]. Our results revealed that SPc could self-assemble with cellobiose with a drug loading content of $12.70 \pm 1.07\%$ (Supplementary Table 1). The cellobiose/SPc complex activated endocytosis in potato cells, evidenced by up-regulation of the EH-domain-binding protein gene (*StEPSIN*), protein-sorting-associated protein 36 gene (*StVPS36*), vesicle-associated membrane protein gene (*StVAMP1-2*, *StVAMP3-1*) and Ras-related in brain gene (*StRab*) by 1.4 to 2.9-fold compared with cellobiose alone (Supplementary Fig. 6a). Furthermore, plant defense genes such as pathogenesis-related protein 1 (*StPR1*), *StWRKY1*, polyphenol oxidase (*StPPO*) and *StPTI5*, induced by cellobiose, were further up-regulated in combination with SPc. The defense-associated transcription factor *StWRKY1* was prominent among the genes tested, being up-regulated by 920-fold (Supplementary Fig. 6b). Two antimicrobial phytoalexins genes related to

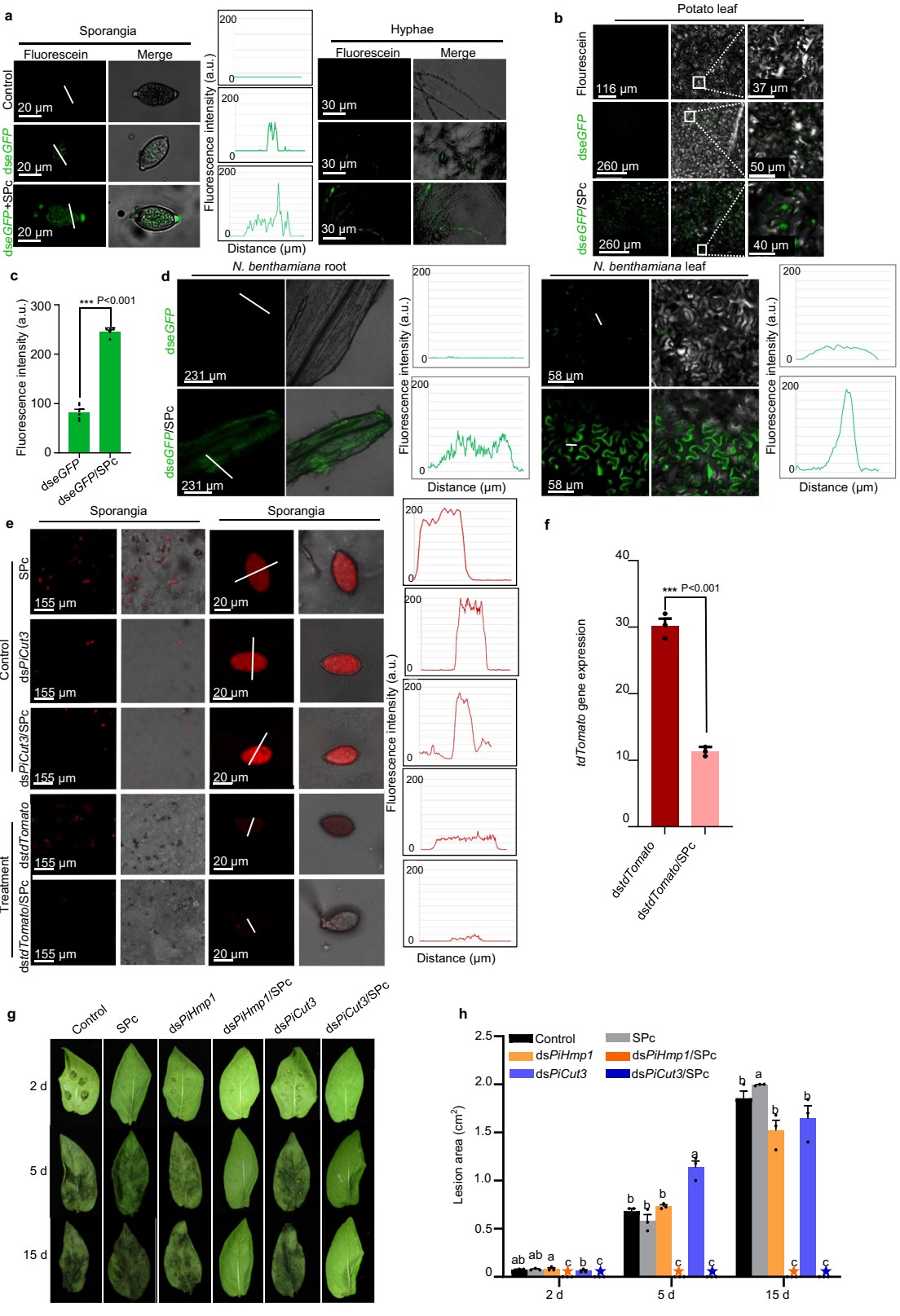

synthesis of coumarin and epicatechin were also up-regulated by 2.5 and 24-fold, and coumarin content was significantly increased by 2.1-fold (Supplementary Fig. 6c–f). SPc alone did not cause up-regulation of immune genes (Supplementary Fig. 7). The above results illustrated that SPc could amplify the cellobiose-induced defense responses by enhancing endocytosis, consistent with reports by Wang et al.[35] and Ma et al.[25].

## Self-assembly mechanism of multicomponent nano-bioprotectant

Nanoparticle-based gene/drug co-delivery systems have shown many advantages since the first application in the cancer therapy field[36]. SPc is a multifunctional nanoparticle that can co-deliver dsRNA and bio-protectant for insect pest control[37]. The current study constructed a multicomponent nano-bioprotectant based on dsRNA and a plant

**Fig. 1 | Enhanced delivery efficiency and protective effect of SPc-loaded dsRNA on leaves. a** SPc promoted the uptake of fluorescently-labelled ds*eGFP* by *P. infestans* sporangia. Five μL of ds*eGFP* (500 ng/μL) and ds*eGFP*/SPc complex were respectively added to both *P. infestans* sporangia and hyphae (12 d) (10⁵ sporangia /mL), and fluorescence intensity was measured at 12 h after the incubation. (*n* = 3 biological replicates). **b, c** SPc promoted the uptake of fluorescent ds*eGFP* by potato leaves after washing (fluorescein was used as the control). **c** The fluorescence intensity was measured at 12 h after the application (*n* = 3 leaves containing 12 inoculation points). The asterisk indicates significant difference according to independent two-tailed *t*-test (\*\*\**p* < 0.001). Bars represent the mean ± SE. **d** SPc promoted the uptake of fluorescently labelled ds*eGFP* by *N. benthamiana* roots and leaves. The plot of the profile (right) indicates that the majority of the green fluorescence (white line) was found in the ds*eGFP*/SPc complex-treated samples. (*n* = 3 biological replicates). **e, f** SPc increased the RNAi efficiency of the *tdTomato* gene in a *P. infestans* transformant expressing tdTomato fluorescent protein. Enlarged images of representative sporangia are shown. Five μL of ds*tdTomato*

(500 ng/μL) and ds*tdTomato*/SPc complex were used, separately, to treat 5 μL of tdT-88069 sporangia suspension (10⁵ sporangia/mL). Both fluorescence intensity and gene expression were examined at 12 h after the treatment. The plot of the profile (right) indicates the majority of the red fluorescence (white line). Bars represent the mean ± SE. Gene expression (**f**) represents the comparison with the non-silenced control. (*n* = 3 biological replicates). The asterisk indicates significant difference according to independent two-tailed *t*-test (\*\*\**p* < 0.001). Bars represent the mean ± SE. **g, h** Enhanced protective effect of SPc-loaded dsRNA on potato leaves. **g** The detached potato leaves were sprayed with various formulations (dsRNA concentration: 500 ng/μL), and sporangia suspension was inoculated onto plant leaves after 24 h. Pictures were acquired at 2, 5 and 15 dpi, and then (**h**) lesion area was measured (*n* = 3 biological replicates). Different letters above each bar indicate significant differences at *p* < 0.05 as determined by one-way ANOVA with Tukey's HSD multiple comparison post hoc test. Bars represent the mean ± SE. A five-pointed star means that the corresponding value is zero. Source data are provided as a Source Data file.

elicitor, which has potential to enhance the antimicrobial bioactivity from two angles for green management of potato late blight. The self-assembly of the multicomponent nano-bioprotectant was investigated using isothermal titration calorimetry (Fig. 2). The cellobiose solution was titrated with SPc aqueous solution to determine their interaction (Supplementary Fig. 8a). The low dissociation constant (Kd) of $4.075 \times 10^{-6}$ M suggested the effective interaction between SPc and cellobiose. The positive values of ΔH and ΔS suggested that non-covalent molecular interactions, such as hydrogen bonds and hydrophobic interactions, performed a dominant role in the self-assembly of the cellobiose/SPc complex. To further investigate the complexation of cellobiose/SPc complex with dsRNA, excessive cellobiose was first incubated with SPc, and then dialyzed to obtain pure cellobiose/SPc complex, which was further titrated with ds*eGFP* aqueous solution. As shown in Fig. 2b, there was an effective, automatic and strong binding force between cellobiose/SPc complex and ds*eGFP*. Meanwhile, values of ΔH and ΔS revealed that the further self-assembly was driven by hydrogen bond and Van der Waals forces. In addition, the agarose gel retardation assay suggested that electrostatic interactions also played an important role in self-assembly of the multicomponent complex (Supplementary Fig. 8b).

The bioactivity of the multicomponent nano-bioprotectant was directly related to its particle size and morphology, which were analyzed using dynamic light scattering and transmission electron microscopy (Fig. 2c–h). The morphology of SPc is approximately spherical with a size of nearly 100 nm (Supplementary Fig. 8c, d). Cellobiose self-aggregated into nearly spherical particles of 219 nm diameter. Its complexation with SPc decreased the particle size to 35 nm, which was similar to previous studies that demonstrated SPc can decrease the particle sizes of various pesticides in aqueous solution[38,39]. Interestingly, the particle size of cellobiose/SPc/ds*eGFP* complex increased to 88 nm due to the adhesion of ds*eGFP* to the surface of the cellobiose/SPc complex. Anthrone-sulfuric acid colorimetry was performed for quantitative analysis of cellobiose content in the multicomponent nano-bioprotectant. The cellobiose content was determined according to the standard curve after 12 h dialysis of cellobiose/SPc/ds*eGFP* complex (Supplementary Fig. 8e). Based on the binding mass ratio of cellobiose/SPc complex to ds*eGFP* in an agarose gel retardation assay, the mass ratio of each component in the multicomponent nano-bioprotectant was 53 (cellobiose): 1000 (SPc): 1000 (ds*eGFP*). The multicomponent nano-bioprotectant could still protect dsRNA from degradation, and deliver dsRNA to *P. infestans* and plant cells (Supplementary Fig. 9).

## Protective effect of multicomponent nano-bioprotectant

To assess the protective effect of the multicomponent nano-bioprotectant on whole plants, disease protection assays were performed under greenhouse and field conditions. In the greenhouse, spraying with the dsRNA/SPc, cellobiose/SPc, and multicomponent

nano-bioprotectant all provided protection against late blight on potato plants (Fig. 3a, b), which was confirmed by *P. infestans* biomass testing results (Fig. 3c). The above three treatments showed smaller amounts of pathogen compared to control and other treatments. There were no marked differences among the above three treatments (Fig. 3c). No defects in the growth of potato plants were observed after any of the treatments. Under field conditions, the seedlings were sprayed with the formulations described above from the second day after the initial spots of late blight were discovered. Due to the suitable climate and susceptible varieties, *P. infestans* infected rapidly and killed nearly all potato foliage in control (water) treatments. By contrast, the multicomponent nano-bioprotectant exhibited the best protective effect with the lowest disease index (20), and its protective effect (68%) was significantly higher than that (53%) of the widely-used mancozeb fungicide at 29 dpi (Fig. 3d–g). Due to the SPc-based nano-delivery system, the multicomponent nano-bioprotectant could not only enter *P. infestans* cells more efficiently for gene silencing, but also enhanced the systemic resistance of plants.

In summary, we applied the SPc-based dsRNA nano-delivery system to perform RNAi in a recalcitrant pathogen, eliminating the dsRNA delivery barrier to broaden the application of RNAi. We co-expressed the dsRNAs of two target genes in a bacterial system to provide a high yield, achieving high production at low cost to meet the demands necessary for field application. We prepared a nano-sized plant elicitor using SPc, amplifying plant defense responses against pathogens. Based on the technologies developed here, we designed a self-assembled multicomponent nano-bioprotectant, realizing sustainable green management of potato late blight via the angles of pathogen inhibition and elevated plant defense (Supplementary Fig. 10). We propose the strategy could be useful for pesticide/drug design and green management of devastating plant diseases and pests.

## Methods

### Synthesis of SPc and dsRNA in vitro

SPc was synthesized according to the method described by Li et al.[27]. In brief, the star initiator Pt-Br was synthesized using commercial pentaerythritol; the polymerization of star initiator with DMAEMA (2-(Dimethylamino) ethyl methacrylate) was carried out under nitrogen atmosphere. The solvent tetrahydrofuran was removed by a rotary evaporator and reused at the end of polymerization to decrease the production cost. Subsequently, dialysis was carried out to purify the crude product in ddH₂O four times, and the final SPc product was obtained as white powder after freeze-drying.

For dsRNA synthesis, total RNA was extracted from *P. infestans* strain T30-4[40] using RNA Simple Total RNA Kit (Tiangen Co., China), and 2 μg of RNA was reverse-transcribed to cDNA using the PrimeScript™ RT Reagent Kit (Takara Co., Japan). Templates of the pathogenesis genes *PiHmp1* (PITG_00375; XP_002908980) and *PiCut3*

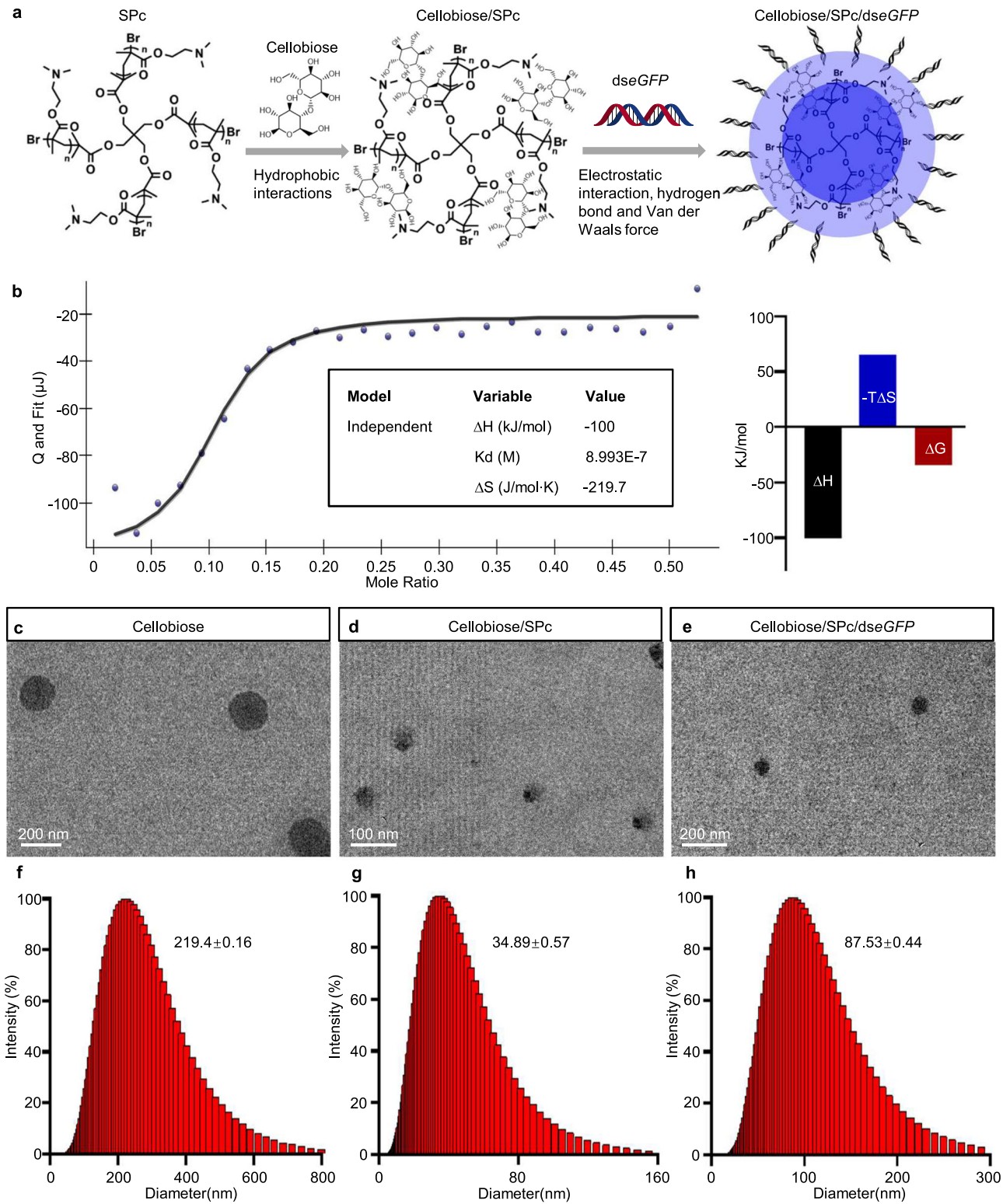

**Fig. 2 | Self-assembly mechanism of cellobiose/SPc/dsRNA complex.**
**a** Schematic illustration of self-assembled cellobiose/SPc/dsRNA complex. **b**, **b-1**
ITC titration of ds*eGFP* solution (0.67 ×10⁻⁶ mol/L) into cellobiose/complex
(1.4 × 10⁻⁶ mol/L). **c–h** TEM images (**c–e**) and particle size distributions (**f–h**) of
cellobiose, cellobiose/SPc complex and cellobiose/SPc/ds*eGFP* complex. The
number indicates the average particle size. Data are from three biological repli-
cates. Source data are provided as a Source Data file.

(PITG_12361; XM_002900240) were amplified using the above cDNA,
cloned into pMD19T-Vector (Takara Co. D102A.) and transformed into
*Escherichia coli* strain DH5α (Tsingke Biotechnology Co., Ltd, China).
The plasmid was extracted and verified by Sanger sequencing, and
then used for dsRNA synthesis using T7 RiboMAX Express RNAi System

(Promega Co., USA). The enhanced green fluorescent protein (eGFP)
gene was amplified by PCR using the template of pB7WGF2 containing
the *eGFP* gene. Fluorescent ds*eGFP* was synthesized using a Fluor-
escein dsRNA Labeling Kit (Roche Co. USA). All primers (Sangon bio-
tech Co. China) are shown in Supplementary Table 2.

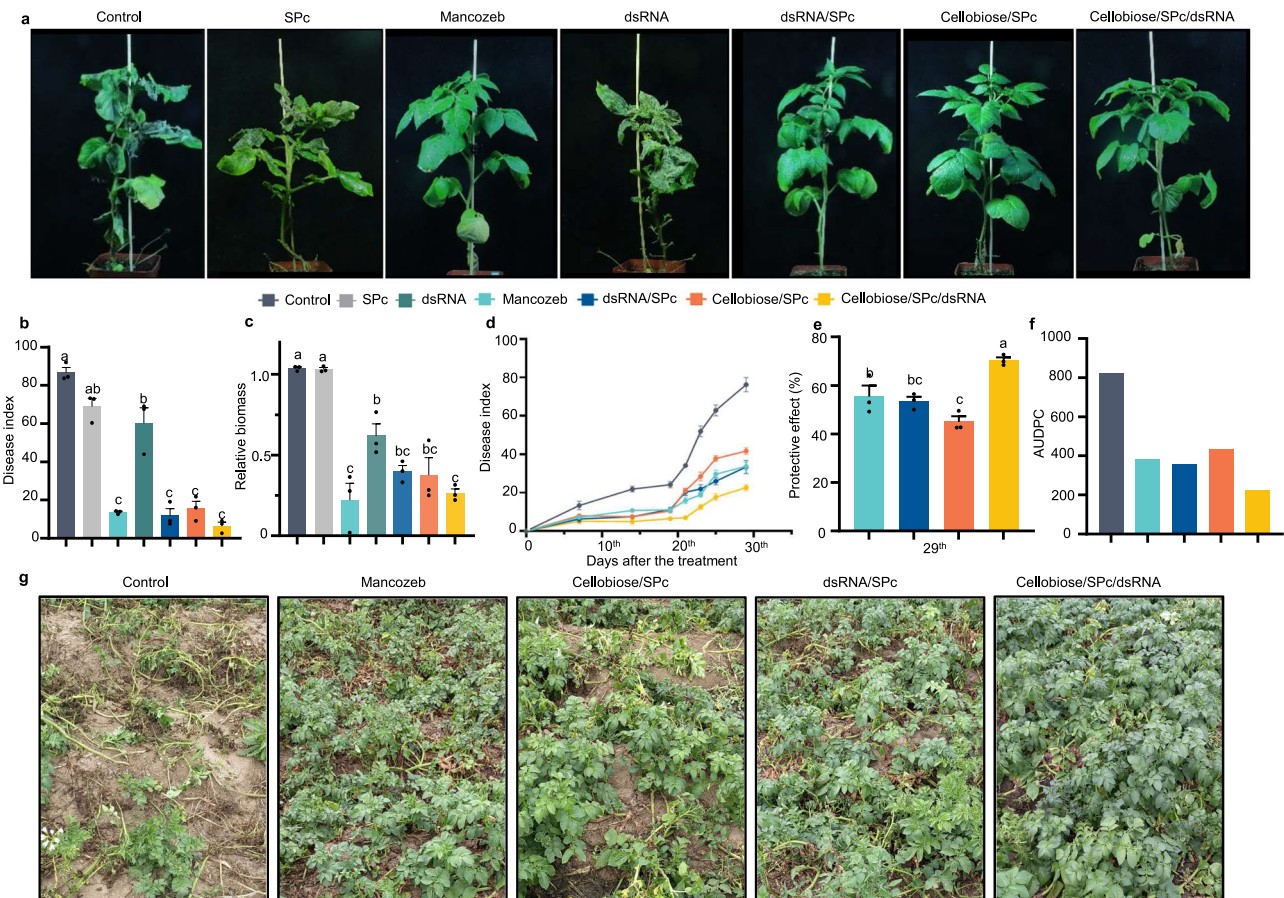

**Fig. 3 | Stronger protective effect of multicomponent nano-bioprotectant toward potted plants in the greenhouse, and under field conditions. a–c** The multicomponent nano-bioprotectant exhibited stronger protective effect on whole plants grown in pots. Various formulations were sprayed onto whole plants (ddH₂O was used as control) and sporangia suspension was then sprayed onto plant leaves after 24 h. **a** Representative photos were acquired at 6 dpi, and then (**b**) disease index and (**c**) relative biomass was quantified ($n = 3$ biological replicates). Different letters above each bar indicate significant differences at $p < 0.05$ as determined by one-way ANOVA with Tukey's HSD multiple comparison post hoc test. Bars represent the mean ± SE. **d-g,** Control efficacy of the multicomponent nano-bioprotectant against potato late blight under field conditions. **d** Disease index was calculated at 7, 14, 19, 21, 23, 25 and 29 day after the initial symptoms of late blight were discovered. Bars represent the mean ± SE. **e** The protective effect was calculated for each treatment at 29 days, analyzed by one-way ANOVA with Tukey's HSD multiple comparison post hoc test. Bars represent the mean ± SE. **f** AUPDC value was calculated after 29 days. The field experiments for each treatment included three random replicate plots. **g** Representative photos were taken at 29 days. Source data are provided as a Source Data file.

To determine whether the fluorescence intensity of ds*eGFP* was changed after its self-assembly with SPc, ELIASA (Molecular Devices) was used to measure the 488-520 nm fluorescence intensity of ds*eGFP* and ds*eGFP*/SPc complex (ds*eGFP* amount was 170 ng). A UV-visible spectrophotometer (Thermo Fisher Scientific) was used to measure the full UV absorption spectra of SPc, ds*eGFP* and ds*eGFP*/SPc complex (final concentration for ds*eGFP*: 20 ng/μL), respectively. Each treatment included three independent samples.

### Stability test of SPc-loaded dsRNA

A gel retardation test was first performed to determine the best mass ratio for the combination of dsRNA with SPc. The ds*PiCut3* (500 ng) was mixed with SPc at various mass ratios, and the mixture (8 μL) was incubated at room temperature for 15 min and then analyzed by agarose gel electrophoresis. To confirm the fast degradation of dsRNA by RNase A, 1 μg of ds*PiCut3* was mixed with different quantities of RNase A (Tsingke Biotechnology Co., Ltd). According to the information provided by the manufacturer (Thermo Co. USA), RNase A can degrade dsRNA under low salt conditions as used in this study. The mixture was incubated for 20 min at 37 °C, and then detected by agarose gel electrophoresis. To investigate the protective effect of SPc on dsRNA, ds*PiCut3* was mixed with SPc at the best mass ratio, and treated with RNase A. The ds*PiCut3*/SPc complex was decomplexed in

0.3% SDS solution and analyzed using agarose gel electrophoresis. Naked ds*PiCut3* was employed as control. The relative band density was determined using ImageJ (National Institutes of Health, USA). Each treatment included three independent samples.

### Uptake efficiency assay of SPc-loaded dsRNA

Two *P. infestans* strains, MZ and tdT-88069, were employed to determine the delivery efficiency of SPc-loaded dsRNA into sporangia and hyphae. The MZ strain was cultured on Rye A agar (1000 mL ddH₂O containing 60 g rye extract, 20 g sucrose and 15 g agar) in darkness at 18 °C for 2 weeks. Five μL of fluorescent ds*eGFP* (500 ng/μL) and ds*eGFP*/SPc complex were respectively mixed with 5 μL of sporangia suspension and hyphae (12 days) (10⁵ sporangia/mL), then dropped inoculated onto Rye A agar, and cultured in darkness at 18 °C for 12 h. Fluorescence images were captured and analyzed using a Leica SP5 confocal microscope (Leica Co., Germany), and the fluorescence intensity was measured using ImageJ software (National Institutes of Health, USA). Each treatment was repeated six times.

Potato and *N. benthamiana* were employed to determine the delivery efficiency of SPc-loaded dsRNA. For the potato uptake assay, 5 μL of 500 ng/μL fluorescent ds*eGFP* and ds*eGFP*/SPc complex were applied separately as droplets on the detached leaves, and fluorescence images were taken at 12 h after application. Similar

experiments were performed with detached leaves and roots of *N. benthamiana*.

The tdT-88069 strain expressing tandem dimer Tomato fluorescent (tdTomato) protein was cultured on Rye A agar containing geneticin (50 mg/L). The ds*tdTomato* (205 bp) was synthesized in vitro. Similarly, five μL of ds*tdTomato* (500 ng/μL), ds*tdTomato*/SPc complex and control (SPc, ds*PiCut3* and ds*PiCut3*/SPc complex) were used to treat 5 μL of tdT-88069 sporangia suspension ($10^5$ sporangia/mL), and red fluorescence was detected with an argon laser at 587 nm and analyzed using the confocal microscope. The silencing efficiency of *tdTomato* was also examined using quantitative real-time PCR (qRT-PCR). Total RNA was extracted from tdT-88069 strain and reverse-transcribed to cDNA. The qRT-PCR was performed with a QuantStudio™ 6 Flex Real-Time PCR System (Applied Biosystems, Thermo Fisher Scientific, USA) using Power SYBR® Green Master Mix (Applied Biosystems). Reactions were performed in triplicate using the following conditions: pre-denaturation at 95°C for 30 s, followed by 35 cycles of denaturation at 95 °C for 5 s, and annealing at 60 °C for 30 s. The *P. infestans actin* gene (XM_002897714.1) transcript was used as an internal control using the $2^{-\Delta\Delta CT}$ method[41]. Each treatment was repeated three times.

### Elevated bioactivity assay of *dsPiHmp1* and *dsPiCut3* with the aid of SPc

Potato cv. *Favorita* was used in the current study due to its high susceptibility to *P. infestans*. Potato plants were grown in a soil-less mix consisting of a 1:1 (v:v) peat-vermiculite mix until 15 cm height in the greenhouse (60–70% RH, 12 h light/day). The detached potato leaves were sprayed with SPc, ds*PiHmp1* (500 ng/μL), ds*PiCut3* (500 ng/μL) and their SPc-loaded complexes with the application amount of 200 μL per leaf; ddH$_2$O was applied as control. After 24 h, 10 μL of sporangial suspension ($10^4$ sporangia/mL) (MZ) was inoculated onto the abaxial side of leaves and incubated in moist sealed boxes at 20 °C with 12 h light/day. The lesion area was measured at 2, 5 and 15 dpi. Each treatment was repeated three times.

### Construction of ds*PiHmp1*+*PiCut3*/SPc complex and its protection assay

ds*PiHmp1* and ds*PiCut3* were co-expressed in vivo for large batch dsRNA production. Two genes were tandemly constructed into pET28a (386595) vector and then co-expressed in *E. coli* BL21(DE3) RNase III- system. A single colony was selected and cultured in 5 mL LB liquid medium supplemented with kanamycin (50 mg/L), followed by overnight shaking at 37 °C. The culture medium was added to new LB medium with kanamycin, and then cultured at 37 °C until the OD$_{600\,nm}$ reached 0.55. The culture incubation was continued for 4 h after adding isopropyl β-D-thiogalactoside (IPTG) (1 mM) for ds*PiHmp1*+*PiCut3* expression. Five mL of LB culture was heated for 20 min at 80 °C, and 75% ethanol added to break the bacterial cell wall. The expressed ds*PiHmp1*+*PiCut3* was dissolved in NaCl solution (150 mM). The obtained ds*PiHmp1*+*PiCut3* was purified by RNA clean kit (Tiangen Co., China) and quantified using NanoDrop2000 spectrophotometer (Thermo Fisher Scientific Co., USA).

The ds*PiHmp1*+*PiCut3* was mixed with SPc at the mass ratio of 1:1 to prepare the ds*PiHmp1*+*PiCut3*/SPc complex. Subsequently, the silencing efficiency in *P. infestans* was examined at 24 h after the treatment using qRT-PCR. The protective effect was further confirmed on detached potato leaves similarly as above. Each treatment was repeated three times.

### Bioactivity assay of cellobiose with the aid of SPc

To investigate the loading capacity of SPc toward cellobiose, 100 mg of SPc and cellobiose were both dissolved in 10 mL of ddH$_2$O and stirred for 2 h. The mixture was dialyzed using regenerated cellulose with a molecular weight cut-off of 1000 Da (Shanghai Yuanye Bio-

Technology Co., China) for 12 h to obtain pure cellobiose/SPc complex. After freeze-drying, product powder was weighed and calculated using the following formula:

$$\text{Drug loading content (DLC, \%)} = \text{weight of drug loaded in complex}/ \\ \text{weight of drug} - \text{loaded complex} \times 100$$

(1)

The experiment was repeated three times.

To determine the immunity response induced by cellobiose in potato, four-week-old potato leaves were sprayed with cellobiose (138 mg/L) and cellobiose/SPc complex prepared according to the DLC using an application volume of 200 μL; ddH$_2$O and SPc (1.815 mg/mL) were applied as controls. RNA was extracted from leaves at 24 h after the spray to examine the expression of cellobiose-induced immunity genes such as *StPR1* (XM_006367028.1), *StPPO* (U22921.1), *StPTI5* (XM_006367134.2), *StWRKY1* (NM_001288675.1), *StCoum* (OL412004.1) and *StEpic* (XM_006349390.2) using qRT-PCR. The relative mRNA levels were normalized to the abundance of a housekeeping gene in potato (Elongation factor 1 alpha, XM_022311019.1) using the $2^{-\Delta\Delta CT}$ method[41]. The endocytosis-related genes *StEPSIN* (XM_006349904.1), *StVPS36* (XM_006347647.2), *StVAMP1-2* (XM_006364202.2), *StVAMP3-1* (XM_049539029.1) and *StRab* (CP055242.1) were also investigated. Each treatment was repeated three times.

As a phytoalexin, coumarin content was further examined using liquid chromatography-tandem mass spectrometry (LC-MS/MS). Coumarin was extracted from 5 g of potato leaves using 50% methanol. After centrifugation, 8 mL of the supernatant was evaporated using a gentle stream of nitrogen, and then dissolved in 200 μL 50% methanol. Samples were separated using an Agilent ZORBAX Eclipse Plus C18 column (2.1 × 100 mm, 3.5 μm) by an Agilent 1260 infinity HPLC system (Agilent Co. USA). The mobile phases consisted of water (A) and acetonitrile (B), both of which contained 0.1% formic acid (v/v). The gradient run was initiated with 5% B for 3 min, then increased to 90% B in the next 20 min linearly at a flow rate of 0.3 mL/min. The HPLC system was coupled online with an Agilent 6520 Q-TOF mass spectrometer equipped with an ESI ion source. The MS parameters were as follow: positive ion mode, gas temperature 325°C, drying gas flow 13 L/min and VCap 4.0 kV. The raw MS data were analyzed by MassHunter (Agilent Co.). Standard curves were obtained from the chromatographic data of standard coumarin with various qualities. Three biological replicates were independently tested for each treatment.

### Preparation and characterization of cellobiose/SPc/dsRNA complex

Excess cellobiose was mixed with SPc at the mass ratio of 1:1 in aqueous solution, and incubated for 15 min at room temperature. The mixture was dialyzed to obtain pure cellobiose/SPc complex. The cellobiose/SPc complex was further incubated with ds*eGFP* at the mass ratio of 1:1 (SPc: ds*eGFP*) and dialyzed for 12 h to prepare cellobiose/SPc/dsRNA complex.

The size and morphology of cellobiose/SPc/dsRNA complexes were investigated using dynamic light scattering (DLS) and transmission electron microscope (TEM), respectively. The particle sizes of SPc (1 mg/mL), cellobiose (1 mg/mL), cellobiose/SPc complex (1 mg/mL) and cellobiose/SPc/ds*eGFP* complex (0.5 mg/mL) were measured in triplicate at 25 °C using DLS (Malvern Instruments Ltd., UK). Their morphologies were further examined using TEM (JEOL-1200, Japan). Five μL of each sample were applied as droplets on the carbon support film, and then air-dried before TEM observation.

### Self-assembly and component quantification of cellobiose/SPc/dsRNA complex

Isothermal titration calorimetry (ITC) can accurately reflect the thermodynamic parameters of the intermolecular interaction between two

substances, and thus suggest the dominant interaction force[42]. One mL cellobiose ($0.138 \times 10^{-3}$ mol/L) was titrated with 500 μL SPc aqueous solution ($0.1 \times 10^{-3}$ mol/L) in a Nano ITC (TA Instruments Waters, USA) to examine the interaction between cellobiose and SPc. ITC titration of ds*eGFP* ($0.67 \times 10^{-6}$ mol/L) into cellobiose/SPc complex ($1.4 \times 10^{-6}$ mol/L) was performed to further illustrate the self-assembly of cellobiose/SPc/dsRNA complex. The heat of interaction during each injection was calculated by the integration of each titration peak using Origin7 software (OriginLab Co., USA). The test temperature was 25 °C, and the ΔG was calculated using the following formula:

$$\Delta G = \Delta H - T\Delta S \qquad (2)$$

Component quantification of the cellobiose/SPc/dsRNA complex was performed using agarose gel retardation assay and anthrone-sulfuric acid colorimetry. Pure cellobiose/SPc complex was mixed with ds*eGFP* at the mass ratio of 3:1, 2:1, 1:1, 1:2 and 1:3 (SPc: ds*eGFP*), and each mixture was analyzed by agarose gel retardation assay. Anthrone-sulfuric acid colorimetry was performed for cellobiose quantification assay. The absorbance of cellobiose at various concentrations was measured at 630 nm after reacting with concentrated sulfuric acid and anthrone to obtain the standard curve. Cellobiose absorbance was measured after the dialysis of cellobiose/SPc/ds*eGFP* complex for another 12 h, and the cellobiose content was calculated.

### Stability test and uptake efficiency assay of multicomponent nano-bioprotectant

To investigate the stability of dsRNA in the multicomponent nano-bioprotectant, cellobiose/SPc/ds*PiHmp1*+*PiCut3* complex (cellobiose: 53 ng; SPc: 1 μg; dsRNA: 1 μg) was treated with RNase A (20 ng), and analyzed using agarose gel electrophoresis. Naked ds*PiHmp1*+*PiCut3* was employed as control. Each treatment included three independent samples.

Multicomponent nano-bioprotectant containing 2.5 μg fluorescent ds*eGFP* was mixed with sporangia suspension, then droplet inoculated on Rye A agar, and cultured in darkness at 18 °C for 12 h. For plant uptake assays, the multicomponent nano-bioprotectant was applied as droplets on the detached leaves, and fluorescence images were taken at 12 h after application and analyzed. Each treatment was repeated six times.

### Protection assay of multicomponent nano- bioprotectant on whole potato plants in greenhouse pots and field

For dsRNA release from the bacteria, lysozyme was added into LB cultures to the final concentration of 1.3 mg/mL for 30 min to break bacterial cell walls, and the mixture was incubated at 75 °C for 5 min to kill the remaining bacteria. The obtained ds*PiHmp1*+*PiCut3* was purified by RNA clean kit (Tiangen Co. China, Cat.#: 4992728) and quantified using a NanoDrop2000 spectrophotometer to prepare the multicomponent nano-bioprotectant. The cellobiose was firstly mixed with SPc according to the component quantification of cellobiose/SPc/dsRNA complex, and then assembled with dsRNA at the final concentration of 24 mg/L (dsRNA).

Four-week-old potato plants were sprayed with the formulations of cellobiose (138 mg/L), cellobiose/SPc complex (cellobiose: 138 mg/L), naked ds*PiHmp1*+*PiCut3* (24 mg/L), ds*PiHmp1*+*PiCut3*/SPc complex (dsRNA: 24 mg/L), cellobiose/SPc/ds*PiHmp1*+*PiCut3* complex (cellobiose: 138 mg/L; SPc: 2.76 g/L; dsRNA: 24 mg/L) as well as controls water and SPc (2.76 g/L) on both sides of the leaf surfaces using an atomizer. All plants were sprayed with *P. infestans* (MZ) sporangia suspension ($1.5 \times 10^4$ sporangia/mL) at 24 h after the treatment and incubated at 20 °C with 60–70% relative humidity, 12 h light/day. Representative infected leaflets were chosen for photography. The relative biomass of *P. infestans*/potato was quantified using qRT-PCR at 6 dpi[43]. Each treatment included three potted plants and was repeated three times.

The field trial was carried out to determine the protective effects of various formulations in a potato field (Chengde, Hebei province, China) where naturally occurring late blight was prevalent due to the suitable climate. The *P. infestans* susceptible *cv.* Wotu 5 was grown. The field trials were at different geographical locations (different fields), sown on separate dates between April 1 to 10. There were three trials of the experiments. Plots were randomly assorted, each treatment contained three field plots, and each field plot was 20 m². Plots were treated with cellobiose/dsRNA/SPc and other treatments on the same dates between July 1-29, since the first late blight disease symptoms were observed at that time. Various formulations were prepared similarly as described above, and sprayed onto the foliage immediately after the first disease symptoms were observed. Formulations included water (control), commercial fungicide Mancozeb (Shuangji Co., China) (local application amount: 5.6 kg/ha), cellobiose/SPc complex, ds*PiHmp1*+*PiCut3*/SPc complex and cellobiose/SPc/ds*PiHmp1*+*PiCut3* complex. Disease records were taken at various time intervals by visually assessing the percent of the blighted leaf area per plant. The leaf disease index and protective effect were assessed using the following formula[44]. Protection effect was assessed by disease index.

$$Disease\ index = \sum \frac{(the\ number\ of\ samples \times Disease\ grade)}{(total\ sample\ size \times Highest\ disease\ class)} \times 100 \qquad (3)$$

$$Percent\ of\ time\ duration\ protection = 1 - \frac{Mean\ treated\ disease\ index}{Control\ isease\ index} \times 100 \qquad (4)$$

Control efficacy of season-long protection for each formulation was calculated as the mean area under disease progress curves (AUDPCs)[45] for all treatment plots. The potato disease classification method was adopted according to the 9 grades classification standard of CIP (International Potato Center) (Supplementary Table 3).

### Statistical analysis

Statistical analysis was conducted using SPSS 19.0 software (SPSS Inc., USA). Independent *t*-test was used to determine the significant differences between two samples. The other data were analyzed using the Tukey HSD test at the *P* value < 0.05 level of significance. The descriptive statistics are shown as the mean value and standard errors of the mean.

### Reporting summary

Further information on research design is available in the Nature Portfolio Reporting Summary linked to this article.

## Data availability

Data supporting the findings of this work are available within the paper and its Supplementary Information files. A reporting summary for this Article is available as a Supplementary Information file. Potato database [http://spuddb.uga.edu/] was used in this study. *P. infestans* (T30-4 strain) genomic data can be accessed at NCBI [https://www.ncbi.nlm.nih.gov/datasets/genome/GCF_000142945.1/]. Source data are provided with this paper.

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

## Acknowledgements

This work was supported by National Key Research and Development Program of the Ministry of Science and Technology (grant numbers: 2022YFD1401800) to D.D., J.S., X.W. and S.Y.; The National Natural Science Foundation of China (grant numbers: 32372558; 32061130211) to D.D. and X.W.; Biotechnology and Biological Sciences Research Council grant BB/S003096/1, European Research Council (ERC)-Advanced grant PathEVome (787764), and Scottish Government Rural and Environment Science and Analytical Services Division (RESAS) to P.R.J.B. and S.C.W. We thank

Associate Prof. Jing Maofeng for reviewing the manuscript and providing suggestions. All authors acknowledge Ministry of Agriculture and Rural Affairs of the People's Republic of China for providing official statistics for potato late blight.

## Author contributions

X.W. and S.Y. designed the project and interpreted the data. J.S. and D.D. oversaw the project. M.Y. provided nanomaterial resources. Y.W., M.L. and J.Y. conducted experiments and analyzed the data. S.C.W. and P.R.J.B. analyzed data and revised manuscript. S.Y. and X.W. wrote the manuscript. All authors edited the manuscript.

## Competing interests

The authors claim no competing interests.
