## [Peer Review File · Nature Communications]

High-efficiency green management of potato late blight by a self-assembled multicomponent nano-bioprotectantReviewers' Comments:

Reviewer #1:

Remarks to the Author:

The manuscript entitled "High-efficiency green management of potato late blight by self-assembled multicomponent nano-bioprotectant" from Wang et al. is an interesting study and research in this field it is needed.

The manuscript presents interesting results and strategies to management of potato late blight.

In the point of view of this review, this manuscript is a regular paper there is not enough novelty to be stated as a Brief Communication in Nature Communication. Maybe could be suitable for a Agri sector journals.

The manuscript has size of a regular papers, easily it is possible to see a lot of results in Supplementary materials and in this way, the submission is not suitable for the selection.

In addition, about the novelty, there were a lot of published papers that works with different strategies to control pests in agriculture. Also, it is important to consider that even with good results presented, the authors should state about the potential use of this technologies in the field, this mean, what about the scale up processes? what about the costs of this system? the system can be very interesting, but taking a look at the methodology, the process is not easy to scale up and the solvent and steps are not green (or sustainable).

In this context, as mentioned before, the manuscript is interesting, but I don't agree that fits with the scope of the journal and also as a Brief Communication, that in my opinion should be present really outstanding processes/results and technologies to contribute to pest control, but thinking in the life cycle, that the system can be sustainable. In summary, I suggest author to submit for a journal more from the Agri field.

Reviewer #2:

Remarks to the Author:

The manuscript by Wang et al. is a very nice and important contribution to the emerging field of RNA drug activity and delivery.

It is clear that the authors have great expertise in nanocarriers and formulation techniques. In addition, the Birch lab has great expertise in Phytophthora pathology.

However, to make it an unassailable contribution, some experimental aspects with respect to the use of dsRNA in plant protection need to be considered in more detail. (1) The important information on the length of the RNA used in the experiments is lacking (at least, I cannot easily find it, suggesting that it should be given in a more prominent place). (2) Information on the concentration of dsRNA used is largely absent from the text. At least in the figure legends, information on concentrations and more experimental details are urgently needed. (3) In reviewing the information given in the MM, it is a critical point that the dsRNA concentration (e.g., per leaf) is very unusually high. This is critical not only with respect to the immunogenic activities of dsRNA.

Specific minor aspects:

I do not find legends to Figures 4 and 5?

L48: To my knowledge, RNase A does not degrade dsRNA?

L50: First mentioning dseGFP; define! The same later on: define dtTomato.

L53: compared to what?

L79: Scale-up: can the authors rule out other E. coli components, e.g. flg15?

Major aspects:

(1) If the reviewer is correct that 2.5 µg of fluorescein-labeled dsRNA was used in the dsRNA uptake experiment, a control must be included for such a large amount of fluorescein to eliminate the possibility that free fluorescein will give these results.

(2) All experiments with exogenous fluorescent dsRNA: An additional control is required: show nuclease treatments (e.g. *Micrococcus dsRNase*) to exclude that the fluorescent dsRNA adheres to the cell walls. This is especially necessary when plant surfaces are shown.

(3) The commercial fungicide (mancozeb) should be applied at much lower concentrations (2 kg/hectare versus 5.6 kg/hectare) and immediately after the first signs of disease symptoms. In the field trial, it was applied 4-5 weeks after disease symptoms, limiting its effectiveness.

(4) If the reviewer is correct, 100 µg dsRNA/leaf was used for the detached leaf test. What amount of dsRNA was used for the field test?

Reviewer #3:

Remarks to the Author:

This is an interesting study. The authors have indeed demonstrated use of a multi-component nanoparticle to suppress potato late blight. I have just a few questions/comments.

It's very difficult for me to know what "high-efficiency" is. Is there some quantitative measure of efficiency? The authors have demonstrated that the use of nanoparticles is more effective (and therefore more efficient?) than not using nanoparticles.

L72. I was intrigued by the claim that the protective effect persisted for 15 days. On what data was that claim based? I could find no place where the persistence of the protective effect was tested. In the greenhouse tests, inoculations occurred 24 h after treatment. Certainly, the protective effect persisted for at least 24 hours. Where were the inoculations that were done at 15 days after treatment?

The description of the field tests did not identify when inoculations occurred. I found it intriguing that there was no disease before treatment, but very soon after treatment there was inoculation. Were the plots artificially inoculated? Please provide more detail. Were there any secondary infections? Additional description of the field test would be helpful. Did the lesions sporulate? What was the variance in the experiment?

L141. There are diverse disease indices for potato late blight. I request that the authors provide a simple narrative description of the disease index they used (more than just a citation). What is the definition of "protective effect"?

W. E. Fry

Reviewer #4:

Remarks to the Author:

The authors present a multicomponent nano-formulation to combat potato blight using dsRNA and immune priming of the plant via cellobiose eliciting. The authors characterize the self-assembly of their nanomaterial and demonstrate its application in the greenhouse and in field tests. The novelty of

the paper comes from combining a bio-elicitor with siRNA to mediate plant defense against an oomycete pathogen. In field trials, the combination therapy yielded effects better than commercial chemical pesticides.

- The authors' rationale behind selecting SPc for this line of experiments is unclear despite their citation of a previous paper. Since the nanoparticle forms the basis of this entire paper, the authors should motivate the use of this nanoparticle as a dsRNA delivery vehicle.

- The flow of the paper lines up with the authors building on a simple model on SPc+dsRNA, then introducing low-cost dsRNA production and finally the addition of cellobiose. While this is a logical progression, this leads to inconsistencies in the assays performed for each NP formulation. The confocal images, for instance, are only presented for SPc+dsRNA, whereas the endocytosis assays and particle characterization (DLS/TEM) are only presented for the complex that includes cellobiose. It is unclear if previous results (like protection from RNase A) still stand with the SPc+dsRNA+cellobiose formulation. It may be more effective to perform the assays on all formulations of the NP and will also make the authors' model presented in the last supplementary figure and their conclusions more convincing.

- The experiment with dseGFP is unclear. First, what is dseGFP? It is not clearly defined. I assume dseGFP refers to a dye labeled dsRNA based on text context but it's not obvious. The authors claim the fluorescence intensity difference between dseGFP and dseGFP/SPc is due to "SPc facilitated uptake" into the Sporangia/Hyphae. Given free dseGFP clearly internalizes in Figure 1 a, it is not obvious that SPc facilitates uptake or simply protects cargo (or both) especially since the experiment was conducted over 12 hours.

- Furthermore, it looks like the Figure 1a fluorescence image for dseGFP was captured at the wrong z-plane thus only part of the fluorescein signal was acquired. If not, it is surprising that we see signal within a subset of the Sporangia instead of a lower amount of signal – the authors should explain why this occurs.

- In Figure 1b, the authors should demonstrate that baseline fluorescence for equal amounts of SPc-loaded dseGFP and free dseGFP is roughly equivalent prior to treating leaves. Otherwise, their intensity analysis does not hold water.

- In Figure 1d, how do the authors select which cells to assess tdTomato expression in? It seems like not all the cells uniformly express tdTomato judging by the field of view presented in the SPc only treated cells.

- It seems remarkable that cellobiose, a small, soluble molecule, is self-assembling into nanoparticles. Have the authors seen any reports in literature of this happening. Otherwise, it is possible that the cellobiose used is contaminated. Also, the authors should include a TEM image and DLS of just SPc.

- The authors should substantiate their selection of genes that determine endocytosis is upregulated – given the complexity of this process, they should run the experiment with a greater array of endocytosis-associated genes, or better yet, run a transcriptomic analysis similar to Wang et al. and Ma et al.

- The authors claim RNAi efficiency of SPc-loaded dsPiHmp1 +PiCut3 produced by the bacterial system was comparable to in vitro synthesized dsRNA – comparing Fig S5 and Fig 1e', this does not seem to be the case. Authors also do not state how many days post-treatment data from Fig S5 is collected.

- Visual disease index scores are subjective and thus prone to high variability. It is difficult to evaluate most of the results in Figure 3. If the authors are going to claim there is a difference between dsRNA/SPc, Cellobiose/SPc, and Cellobiose/SPc/dsRNA treatments, more explanation of the methodology for evaluating disease index scores is needed (rather than simply referencing a previous publication).

- A quantitative PCR-based assay quantifying the relative amount of pathogen:plant DNA with marker gene(s) would support the authors' conclusions in Figure 3, where the differences between the NP formulations are not as visually obvious and are fairly marginal on the disease index measures. Furthermore, the controls seem inconsistent in Figure 3 between the greenhouse and field trials – they should show cellobiose, 'naked' dsRNA, and Mancozeb in both the experiments in Figure 3. While the authors land on the conclusion that cellobiose/SPc/dsRNA NP is the superior version, I don't think this conclusion is well-supported without quantifiable data and better controls.

Minor Comments

- The a', b' labeling scheme is difficult to follow. Also, please avoid putting the letters out of alphabetical order for the figure panels so they are easier to read.
- It would be helpful to see all the individual data points comprising the bar graphs plotted.
- The authors mention n=3 biological replicates. What is the nature of these replicates? Or do they mean biological repeats or experimental repeats? Otherwise, it appears as though each experiment as conducted once.
- Plots in Figure 1 of Fluorescence intensity are missing labels and units on the x-axis
- The sporangia in Figure 1d treated with dstdTomator/SPc looks lysed.
- The data and discussion on self-assembly of cellobiose and SPc are not essential to the authors' main claims and can be placed in the supplement.
- K_a is simply the inverse of K_d – the authors can just report K_d as it is the more frequently encountered number for binding affinity strength.
- Methods, line 67: It is not clear how the RNA is purified during this process. Is total RNA from the lysed E. coli being precipitated and used for experiments?
- SF 1: Figure may not be necessary.
- SF 2: Please expand on what SPc nanoparticles are, and the conclusion of the mass ratio gel.

RESPONSE TO REVIEWERS' COMMENTS

First of all, we want to thank the Nature Communications editors for granting us the opportunity to improve our manuscript. At the same time, we would like to thank all reviewers for their valuable comments and suggestions, which made our article more complete and containing more detailed data. We also appreciate the timeframe to achieve our revision with the primary goal of reaching the high-quality standards expected in this journal.

- **Reviewer 1**

The manuscript entitled "High-efficiency green management of potato late blight by self-assembled multicomponent nano-bioprotectant" from *Wang et al.* is an interesting study and research in this field it is needed.

We appreciate the encouraging comments of the reviewer. We have considered all detailed suggestions in preparing our revised manuscript. We hope that our responses will be deemed satisfactory.

Query 1. In the point of view of this review, this manuscript is a regular paper there is not enough novelty to be stated as a Brief Communication in Nature Communication. Maybe could be suitable for a Agri sector journals.

This article highlights three novel technical innovations: 1) the successful nanocarrier-based dsRNA delivery, overcoming the bottleneck of *Phytophthora* RNAi; 2) the low-cost large-scale production of dsRNA via optimized, engineered bacteria; and 3) the development and application of the first multicomponent nano-bioprotectant targeting plant diseases. We have illustrated these novelties in **Lines 156-165** in the manuscript. This work is situated at the forefront of interdisciplinary research, encompassing multiple fields including biology, materials science, and chemistry. The writing style is geared towards popular science audiences and has broad appeal. Accordingly, submission to leading comprehensive journals like Nature Communications would be most appropriate in our opinion. Purely agricultural or biological journals may not fully capture the innovative contributions presented in this article.

Query 2. Also, it is important to consider that even with good results presented, the authors should state about the potential use of this technologies in the field, this mean, what about the scale up processes?

The nanomaterial utilized in this experiment can be synthesized in bulk via an eco-friendly, two-step process at a low cost of about \$50 US per kilogram. This manufacturing approach has been successfully upscaled to an industrial level. The biosynthesis of dsRNA in the engineered bacteria is a conventional process; *Escherichia coli* fermentation, which is more cost-effective than *Bacillus thuringiensis* and *Bacillus subtilis* commonly used in the market, incurring a production cost of \$1 US per gram. Cellobiose that is commercially produced by several pesticide companies, is cost-controllable. Consequently, we have developed a prototype of a multicomponent nano-bioprotectant that will be applied approximately 60 hectares in this year. We have already applied for Chinese Patent No. 202310191193.2 and plan to seek pesticide registration.

Query 3. What about the costs of this system?

Thanks for your input. Many companies such as Bayer and Greenlight Biosciences have developed dsRNA pesticides targeting plant diseases and insect pests, and applied for pesticide registration. Regarding the cost of our system, we have analyzed the production cost that includes SPc nanomaterial (\$50 US per kilogram), dsRNA (\$1 US per gram), and cellobiose (\$0.9 US per kilogram), with a total application cost of \$162.12 US per hectare.

Query 4. The system can be very interesting, but taking a look at the methodology, the process is not easy to scale up and the solvent and steps are not green (or sustainable). In my opinion should be present really outstanding processes/results and technologies to contribute to pest control, but thinking in the life cycle, that the system can be sustainable.

We greatly appreciate your valuable feedback, and we would like to emphasize that the design of our crop protectant was implemented using an eco-friendly method. Firstly, our dsRNA is biosynthesized using an engineered strain of *E. coli*, which poses no environmental threat. Secondly, the SPc nanocarrier exhibits low toxicity to environmental organisms and can be easily degraded in the environment. Furthermore, the solvents applied in SPc synthesis can be recycled and reused. Thus, the production process of SPc is eco-friendly. Additionally, the plant resistance elicitor cellobiose derived from plant cell walls is safe for application. Our multicomponent nano-bioprotectant formulation does not contain any organic additives. Compared to conventional chemical pesticides, the construction of multicomponent nano-bioprotectant in this study is eco-

friendly and sustainable.

Query 5. In this context, as mentioned before, the manuscript is interesting, but I don't agree that fits with the scope of the journal and also as a Brief Communication.

This work is situated at the forefront of interdisciplinary research, encompassing multiple fields including biology, material science and chemistry. The writing style is geared towards popular science audiences and has broad appeal. Accordingly, submission to leading comprehensive journals like Nature Communications would be most appropriate in our opinion. Purely agricultural or biological journals may not fully capture the innovative contributions presented in this article.

- **Reviewer 2**

The manuscript by Wang et al. is a very nice and important contribution to the emerging field of RNA drug activity and delivery. It is clear that the authors have great expertise in nanocarriers and formulation techniques. In addition, the Birch lab has great expertise in *Phytophthora* pathology. We are glad that the reviewer found our study important and appreciate your encouraging comments.

Query 1. The important information on the length of the RNA used in the experiments is lacking (at least, I cannot easily find it, suggesting that it should be given in a more prominent place).

Done. Please see **Line 73** in the revised manuscript, and **Line 47** in methods.

Query 2. Information on the concentration of dsRNA used is largely absent from the text. At least in the figure legends, information on concentrations and more experimental details are urgently needed.

Apologies. We have added the concentration of dsRNA into the figure legends, see **Lines 230-232, 240-241** and **246-247** in the revised manuscript, and **Lines 37-38** and **47-49** in the methods.

Query 3. In reviewing the information given in the MM, it is a critical point that the dsRNA concentration (e.g., per leaf) is very unusually high. This is critical not only with respect to the immunogenic activities of dsRNA.

To evaluate the protective effect of dsRNA against late blight, each leaf was treated by spraying 200 μL of 500 $\text{ng}/\mu\text{L}$ synthesized dsRNA (**Lines 62-63** in methods), which is lower than previous studies: *"Three real leaves from each plant were treated by spraying approximately 5 mL of a suspension of 100 $\text{ng}/\mu\text{L}$ dsRNA or BioClay."* (Niño et al., 2022).

The initial results of the *in vitro* leaf protection experiment demonstrated significant efficacy for ds*PiHmpl* and ds*PiCut3*. Upon confirmation of its effectiveness, we applied crude bacterial extracts (dsRNA concentration: 24 ng/μL, lower than the kit synthesis) through indoor and field spraying methods. Despite the lower concentration, the application still effectively controlled pathogens.

Reference:

[1] Niño-Sánchez, J. et al. BioClay™ prolongs RNA interference-mediated crop protection against *Botrytis cinerea*. *J. Integr. Plant Biol.* **64**: 2187– 2198 (2022).

Query 4. I do not find legends to Figures 4 and 5?

According to the requirements, Brief Communication (NC journal) can only include three main figures, and the rest of the remaining content and supporting evidence are in the supplementary materials.

Query 5. L48: To my knowledge, RNase A does not degrade dsRNA?

Our results show that RNase A can degrade dsRNA, and previous work has similar results:

[1] Wang, Z. et al. Functionalized carbon dot-delivered RNA nano fungicides as superior tools to control phytophthora pathogens through plant rdrp1 mediated spray-induced gene silencing *Adv. Funct. Mater.* 2213143 (2023)

[2] Ma, Z. et al. Visualization of the process of a nanocarrier-mediated gene delivery: stabilization, endocytosis and endosomal escape of genes for intracellular spreading. *J. Nanobiotechnol* **20**, 124 (2022).

Query 6. First mentioning *dseGFP*; define! The same later on: define *tdTomato*.

Done. Please see **Lines 52-53** and **60** in the revised manuscript.

Query 7. L53: compared to what?

The naked *dseGFP* was used as the control. Please see **Lines 57-58** in the manuscript.

Query 8. L79: Scale-up: can the authors rule out other *E. coli* components, e.g. flg15?

In the process of releasing dsRNA from crushing the engineered bacteria, we performed high temperature treatment to ensure that flg15 and other components are inactivated. Our empty vector control proved that other components did not induce plant resistance (Supplementary Fig. 5).

Query 9. If the reviewer is correct that 2.5 μg of fluorescein-labeled dsRNA was used in the dsRNA

uptake experiment, a control must be included for such a large amount of fluorescein to eliminate the possibility that free fluorescein will give these results.

Thank you for your suggestion. We have added a proper fluorescein control. See modified Fig. 1

Query 10. All experiments with exogenous fluorescent dsRNA: An additional control is required: show nuclease treatments (e.g. *Micrococcus* dsRNase) to exclude that the fluorescent dsRNA adheres to the cell walls. This is especially necessary when plant surfaces are shown.

S_Pc strongly protects dsRNA from RNase degradation (Supplementary Fig. 1), so our method can only be carried out by washes with water to remove dsRNA that has not been taken up by cells for three times. The dsRNA delivery by nanomaterials in cells, plants, fungi and oomycetes has been confirmed in the following studies.

Reference:

[1] Qiao, L. et al. Spray-induced gene silencing for disease control is dependent on the efficiency of pathogen RNA uptake. *Plant Biotechnol. J.* **19**, 1756–1768 (2021).

[2] Ma, Z. et al. Visualization of the process of a nanocarrier-mediated gene delivery: stabilization, endocytosis and endosomal escape of genes for intracellular spreading. *Journal of Nanobiotechnology*, **20**: 124 (2022).

Query 11. The commercial fungicide (mancozeb) should be applied at much lower concentrations (2 kg/hectare versus 5.6 kg/hectare) and immediately after the first signs of disease symptoms. In the field trial, it was applied 4-5 weeks after disease symptoms, limiting its effectiveness.

Indeed, the applied amount of mancozeb (5.6 kg/hectare) in our study was higher than 2 kg/hectare. We considered the control pressure of late blight is high, and the local farmers commonly use mancozeb at higher concentration than recommendation, that's why we applied this concentration. In the field trial, we applied fungicides immediately after the first signs of disease symptoms, but not 4-5 weeks to control, we have modified the language description (**Lines 169-170** in methods).

Query 12. If the reviewer is correct, 100 µg dsRNA/leaf was used for the detached leaf test. What amount of dsRNA was used for the field test?

The 100 µg dsRNA/leaf was used for the detached leaf test, and 24 g dsRNA/hectare was used in the field trail (**Line 156** in methods). These details have now been included.

● **Reviewer 3**

This is an interesting study. The authors have indeed demonstrated use of a multi-component

nanoparticle to suppress potato late blight. I have just a few questions/comments.

We appreciate the encouraging comments of the reviewer.

Query 1. It's very difficult for me to know what "high-efficiency" is. Is there some quantitative measure of efficiency? The authors have demonstrated that the use of nanoparticles is more effective (and therefore more efficient?) than not using nanoparticles.

Compared to naked dsRNA, the SPc-loaded dsRNA exhibited stronger stability and higher delivery efficiency. Compared to cellobiose alone, the SPc-loaded cellobiose induced stronger plant defense. The multicomponent nano-bioprotectant showed the strongest protective effect among various formulations in field trial. Thus, "high-efficiency" was used in many places.

Query 2. L72. I was intrigued by the claim that the protective effect persisted for 15 days. On what data was that claim based? I could find no place where the persistence of the protective effect was tested. In the greenhouse tests, inoculations occurred 24 h after treatment. Certainly, the protective effect persisted for at least 24 hours. Where were the inoculations that were done at 15 days after treatment?

Sorry for the misunderstanding. We focused on the lesion area at 15 dpi, not the persistence of the protective effect. We have modified this description: "there were still no disease symptoms in the leaves after 15 days of initial inoculation." (**Lines 80-81** in manuscript). Please see Fig. 1g and h.

Query 3. The description of the field tests did not identify when inoculations occurred. I found it intriguing that there was no disease before treatment, but very soon after treatment there was inoculation. Were the plots artificially inoculated? Please provide more detail. Were there any secondary infections? Additional description of the field test would be helpful. Did the lesions sporulate? What was the variance in the experiment?

The field experiments in this study were carried out in late blight-prone areas and infected varieties were planted at the same time, so late blight could occur naturally in this plot. In this study, the crop protectant was applied from the second day after the initial spots of late blight were discovered. At the same time, microscopic examination showed sporogenesis of *Phytophthora infestans*; We have added more details in the methods (**Lines 148-150** in the revised manuscript). Commonly, there are secondary infections for many times especially in high-pressure area of late blight occurrence naturally. On the other hand, the influence of natural variables is relatively small: the terrain of the experimental plots we selected is relatively flat, the occurrence of late blight is uniform, and the plots with different treatments are completely randomly distributed. The content has been

supplemented in methods (Lines 165-166).

Query 4. L141. There are diverse disease indices for potato late blight. I request that the authors provide a simple narrative description of the disease index they used (more than just a citation). What is the definition of “protective effect”?

Thanks for your suggestion. We have added more description of the disease index (Line 172-176 and 177-179 in methods). Protection effect was assessed by disease index, which was calculated with the formula of percent of time-duration protection = $1 - (\text{Mean disease index-treated} / \text{Disease index-control}) \times 100$ (Cohen et al., 2018). The content has been supplemented in methods (lines 169-170).

Reference:

[1] Cohen, Y. et al. Oxathiapiprolin-based fungicides provide enhanced control of tomato late blight induced by mefenoxam-insensitive *Phytophthora infestans*. *PLoS one* **13**, e0204523 (2018).

● Reviewer 4

The authors present a multicomponent nano-formulation to combat potato blight using dsRNA and immune priming of the plant via cellobiose eliciting. The authors characterize the self-assembly of their nanomaterial and demonstrate its application in the greenhouse and in field tests. The novelty of the paper comes from combining a bio-elicitor with siRNA to mediate plant defense against an oomycete pathogen. In field trials, the combination therapy yielded effects better than commercial chemical pesticides.

We thank the reviewer for highlighting the important aspects of our study.

Query 1. The authors’ rationale behind selecting SPc for this line of experiments is unclear despite their citation of a previous paper. Since the nanoparticle forms the basis of this entire paper, the authors should motivate the use of this nanoparticle as a dsRNA delivery vehicle.

The current study provides clear elucidation of the mechanism underlying SPc-mediated dsRNA delivery, revealing superior protective and delivery efficacy. Furthermore, its low synthetic cost makes it amenable to large-scale promotion and application. We have added more information about the co-delivery function of SPc (Lines 45-49 in the revised manuscript).

Query 2. The flow of the paper lines up with the authors building on a simple model on SPc+dsRNA, then introducing low-cost dsRNA production and finally the addition of cellobiose. While this is a logical progression, this leads to inconsistencies in the assays performed for each NP formulation.

The confocal images, for instance, are only presented for SPc+dsRNA, whereas the endocytosis assays and particle characterization (DLS/TEM) are only presented for the complex that includes cellobiose. It is unclear if previous results (like protection from RNase A) still stand with the SPc+dsRNA+cellobiose formulation. It may be more effective to perform the assays on all formulations of the NP and will also make the authors' model presented in the last supplementary figure and their conclusions more convincing.

Thanks for your advice. According to your suggestion, we have added another attached figure (Supplementary Fig. 9), which mainly includes the protective effect of multicomponent complex on dsRNA, as well as the delivery of multicomponent nano-bioprotectant on *Phytophthora infestans* spores, mycelia and plants. We also revised the manuscript (**Lines 143-144**) and methods (**Lines 138-146**).

Query 3. The experiment with *dseGFP* is unclear. First, what is *dseGFP*? It is not clearly defined. I assume *dseGFP* refers to a dye labeled dsRNA based on text context but it's not obvious. The authors claim the fluorescence intensity difference between *dseGFP* and *dseGFP/SPc* is due to "SPc facilitated uptake" into the Sporangia/Hyphae. Given free *dseGFP* clearly internalizes in Figure 1 a, it is not obvious that SPc facilitates uptake or simply protects cargo (or both) especially since the experiment was conducted over 12 hours.

The *dseGFP* is the dye labeled dsRNA targeting *enhanced green fluorescent protein* gene. The relevant content description has been supplemented (**Lines 52-53** in the revised manuscript).

The SPc can highly protect dsRNA from RNA enzyme degradation and promote the dsRNA delivery (Supplementary Fig. 1), and the elevated uptake might be due to the combination of these two aspects. We have revised the manuscript (**Lines 64-66**). These references below also supported the conclusion.

Reference:

- [1] Ma, Z. et al. Visualization of the process of a nanocarrier-mediated gene delivery: stabilization, endocytosis and endosomal escape of genes for intracellular spreading. *Journal of Nanobiotechnology* **20**: 124 (2022)
- [2]. Li, M. et al. A gene and drug co-delivery application helps to solve the short life disadvantage of RNA drug. *Nano Today* **43**, 101452 (2022).
- [3] Wang, Y. et al. Nanoparticle carriers enhance RNA stability and uptake efficiency and prolong the protection against *Rhizoctonia solani*. *Phytopathol. Res.* **5**: 2. (2023)
- [4] Yan, S. et al. Spray method application of transdermal dsRNA delivery system for efficient gene silencing and pest control on soybean aphid *Aphis glycines*. *J. Pest Sci.* 93:449-459 (2020)

[5] Ma, Z. et al. A first greenhouse application of bacteria-expressed and nanocarrier-delivered RNA pesticide for *Myzus persicae* control. *J. Pest Sci.* 96: 181-193 (2023)

Query 4. Furthermore, it looks like the Figure 1a fluorescence image for *dseGFP* was captured at the wrong z-plane thus only part of the fluorescein signal was acquired. If not, it is surprising that we see signal within a subset of the Sporangia instead of a lower amount of signal – the authors should explain why this occurs.

Thanks for your suggestion. We reviewed each scanning layer of the Z-plane and found that the fluorescence signal of dsRNA alone was weaker than that of the complex. We have modified Fig. 1a.

Query 5. In Figure 1b, the authors should demonstrate that baseline fluorescence for equal amounts of SPc-loaded *dseGFP* and free *dseGFP* is roughly equivalent prior to treating leaves. Otherwise, their intensity analysis does not hold water.

We performed additional experiments to confirm that the combination of dsRNA and SPc did not change the UV absorption spectra of dsRNA. Meanwhile the fluorescence intensity of SPc-loaded *dseGFP* is similar to that of free *dseGFP*. The relevant content description has been added in Supplementary Fig.2 (**Lines 53-54** in the revised manuscript, and **Lines 18-23** in methods).

Query 6. In Figure 1d, how do the authors select which cells to assess tdTomato expression in? It seems like not all the cells uniformly express tdTomato judging by the field of view presented in the SPc only treated cells.

We analyzed the *tdTomato* gene expression level of the collected mycelium treated by *dstdTomato* to illustrate the efficient delivery of *dstdTomato* by SPc (Fig. 1f in the revised manuscript). It is difficult to uniform all of the cells in tdTomato strain, because not all of the sporangia could express red fluorescence in the transgenic strain. We chose some representative sporangia with average fluorescence intensity to show the difference among various treatments (Fig. 1e).

Query 7. It seems remarkable that cellobiose, a small, soluble molecule, is self-assembling into nanoparticles. Have the authors seen any reports in literature of this happening. Otherwise, it is possible that the cellobiose used is contaminated. Also, the authors should include a TEM image and DLS of just SPc.

Thank you for your suggestion. In our previous study, there are many pesticides such as thiamethoxam, osthole and dinotefuran which can self-assemble into particles ranging from 269 to

575 nm in solvent. The related references are shown below. The applied cellobiose is not contaminated. We have provided the TEM image and DLS of SPc. Please see **Lines 133-134** in the revised manuscript, **Lines 117** and **119-120** in methods, and supplementary Fig. 8.

Reference:

[1] Yan, S. et al. Nanometerization of thiamethoxam by a cationic star polymer nanocarrier efficiently enhances the contact and plant-uptake dependent stomach toxicity against green peach aphids. *Pest Manag. Sci.* **77**: 1954-1962 (2021)

[2] Yan, S. et al. Simple osthole/nanocarrier pesticide efficiently controls both pests and diseases fulfilling the need of green production of strawberry. *ACS Appl. Mater. Interfaces*, **13**: 36350-36360 (2021)

[3] Jiang, Q. et al. A nanocarrier pesticide delivery system with promising benefits in the case of dinotefuran: strikingly enhanced bioactivity and reduced pesticide residue. *Environ. Sci.: Nano*, **9**: 988-999 (2022)

Query 8. The authors should substantiate their selection of genes that determine endocytosis is upregulated – given the complexity of this process, they should run the experiment with a greater array of endocytosis-associated genes, or better yet, run a transcriptomic analysis similar to Wang et al. and Ma et al.

The main reason why we did not conduct transcriptome determination was that some genes with uncertain functions were separated out from transcription components. Therefore, we detected the expression of key genes involved in endocytosis and exocytosis directly to illustrate the enhanced endocytosis and exocytosis with the help of SPc. We have tested more endocytosis and exocytosis related genes in Supplementary Fig. 6a and Supplementary Table 2; (**Lines 101-105** in the revised manuscript, and **Lines 95-97** in methods).

Query 9. The authors claim RNAi efficiency of SPc-loaded ds*PiHmp1 +PiCut3* produced by the bacterial system was comparable to in vitro synthesized dsRNA – comparing Fig S5 and Fig 1e', this does not seem to be the case. Authors also do not state how many days post-treatment data from Fig S5 is collected.

The graph depicted in Fig. 1e' represents the statistical analysis of lesion area, rather than the transcript knockdown effect of dsRNA. We present the gene silencing efficiency obtained through chemical synthesis of dsRNA in Supplementary Fig. 4, which is comparable to the interference effect of dsRNA produced by engineered bacteria (Supplementary Fig. 5c). For further details regarding the experimental methods and sampling time for RNAi, please refer to the methods

section (Lines 79-80 in methods).

Query 10. Visual disease index scores are subjective and thus prone to high variability. It is difficult to evaluate most of the results in Figure 3. If the authors are going to claim there is a difference between dsRNA/SPc, Cellobiose/SPc, and Cellobiose/SPc/dsRNA treatments, more explanation of the methodology for evaluating disease index scores is needed (rather than simply referencing a previous publication).

Thanks for your suggestion. We have supplemented the disease grade classification criteria according to your advice. We added more description about potato disease classification method adopting to the 9 grades classification standard of CIP (International Potato Center), and Disease index conception (Lines 177-179 in methods).

Query 11. A quantitative PCR-based assay quantifying the relative amount of pathogen:plant DNA with marker gene(s) would support the authors' conclusions in Figure 3, where the differences between the NP formulations are not as visually obvious and are fairly marginal on the disease index measures. Furthermore, the controls seem inconsistent in Figure 3 between the greenhouse and field trials – they should show cellobiose, 'naked' dsRNA, and Mancozeb in both the experiments in Figure 3. While the authors land on the conclusion that cellobiose/Spc/dsRNA NP is the superior version, I don't think this conclusion is well-supported without quantifiable data and better controls. According to your valuable suggestion, we provided the biomass (the relative amount of pathogen : plant DNA with marker genes) data, and added the mancozeb treatment in Fig. 3 (Lines 147-148 in the revised manuscript, Lines 161-162 in methods). According to the results from the greenhouse experiment, we only carried forward the treatments that showed control of late blight into the field trial. That is, treatments such as naked dsRNA and SPc were not tested under field conditions, as all testing of these treatments until this point had shown no effect in controlling late blight. The primary goal of the field trial was demonstrating that the nanobioprotectant could control late blight under field conditions.

Query 12. The a', b' labeling scheme is difficult to follow. Also, please avoid putting the letters out of alphabetical order for the figure panels so they are easier to read.

Done. We have replaced a' and b' in the article and adjusted the order of the pictures.

Query 13. It would be helpful to see all the individual data points comprising the bar graphs plotted.

Done. The individual data points have been added to the corresponding picture.

Query 14. The authors mention $n=3$ biological replicates. What is the nature of these replicates? Or do they mean biological repeats or experimental repeats? Otherwise, it appears as though each experiment as conducted once.

Done. All of our repeats refer to biological repeats and the expression in the paper has been revised (Lines 244, 249, 257 and 264 in the revised manuscript).

Query 15. Plots in Figure 1 of Fluorescence intensity are missing labels and units on the x-axis
We have marked the X-axis in Figure 1.

Query 16. The sporangia in Figure 1d treated with dstdTomator/SPc looks lysed.

Sorry for the misunderstanding. The spores in Fig. 1d are not lysed. To avoid this misunderstanding, we have replaced the picture.

Query 17. The data and discussion on self-assembly of cellobiose and SPc are not essential to the authors' main claims and can be placed in the supplement.

Done. We have reduced the emphasis on the description of the self-assembly of cellobiose and SPc, and transferred the data to supplementary Fig. 8. See Lines 121-124 in the revised manuscript.

Query 18. K_a is simply the inverse of K_d – the authors can just report K_d as it is the more frequently encountered number for binding affinity strength.

According to your suggestion, we have revised the contents related to K_a in the manuscript. See Lines 121-124.

Query 19. Methods, line 67: It is not clear how the RNA is purified during this process. Is total RNA from the lysed *E. coli* being precipitated and used for experiments?

We have provided the dsRNA purification method (Lines 75-77 in methods). The purified total RNA was predominantly composed of the target dsRNA, which was subsequently used in the experiments.

Query 20. SF 1: Figure may not be necessary.

According to your suggestion, we have deleted Supplementary Fig. 1.

Query 21. SF 2: Please expand on what SPc nanoparticles are, and the conclusion of the mass ratio

gel.

We have revised the manuscript to illustrate the structure of SPc (**Lines 45-49** in the revised manuscript). The tertiary amine of SPc can combine with negatively-charged dsRNA through electrostatic adsorption. Thus, dsRNA loses its electrical charge and gets trapped within gel wells during electrophoresis. Upon complexation of dsRNA with SPc at the mass ratio of 2:1, some unbound dsRNA can be detected by electrophoresis, indicating that the mass ratio of 1:1 between SPc and dsRNA is the optimal combination.

Reviewers' Comments:

Reviewer #2:

Remarks to the Author:

The manuscript was improved in line with the reviewer's suggestions. In particular, my part of the criticisms was fulfilled.

There is one small point to consider: It is common biochemical knowledge that RNA A does not digest dsRNA. If the authors argue that this has been published in two papers (which they also cite), it may already have been misplaced in those papers! I suggest that this point be critically reviewed so that incorrect knowledge is not multiplied by their manuscript.

Reviewer #3:

Remarks to the Author:

I am confused about the number of times and the number of locations in which the field experiment(s) was (were) conducted. This confusion results from reading the manuscript and in the response to my initial query. Please clarify. I strongly suspect there was one trial of the experiment. Please clarify.

Reviewer #4:

Remarks to the Author:

The authors have done much to assuage this reviewer's concerns. A few remain:

- The authors claim that the Spc/Cellobiose/dsRNA is significantly more effective than other treatments at controlling disease. Figure 3 a,c,d,e,f seem to indicate that dsRNA/SPc, Cellobiose/SPc, and Cellobiose/SPc/dsRNA all control disease about equally. N=3 replicates is quite small for a plant disease experiment so I am not convinced the difference between these three treatments is significant. I suggest change the wording to reflect that all three behave similarly.

- Do the authors observe any detrimental impacts to yield from their treatment? Immune priming is no longer widely pursued commercially due to the trade-off between growth and immunity. If this is to be a commercially viable/relevant finding, this issue should be addressed.

RESPONSE TO REVIEWERS' COMMENTS

Thanks to the Nature Communications' editors for granting us the opportunity to improve our manuscript. We would like to thank all reviewers for their valuable comments to improve our study. We also appreciate the timeframe to achieve our revision with the primary goal of reaching the high-quality standards expected in this journal. We have tried our best to conduct revisions, please see the line-by-line Response to Reviewers' comments for the details.

● Reviewer #2 (Remarks to the Author)

The manuscript was improved in line with the reviewer's suggestions. In particular, my part of the criticisms was fulfilled.

Query 1. There is one small point to consider: It is common biochemical knowledge that RNA A does not digest dsRNA. If the authors argue that this has been published in two papers (which they also cite), it may already have been misplaced in those papers! I suggest that this point be critically reviewed so that incorrect knowledge is not multiplied by their manuscript.

Thanks for your suggestion. After reviewing the information and contacting the companies of related products, we further confirmed once again that RNase A could degrade dsRNA through literature, product manuals and our experimental data (**Supplementary Figure 1**).

RNase A cleaves single-stranded and double-stranded RNA as well the RNA strand in RNA-DNA hybrids. The enzyme is active under a wide range of reaction conditions. Following the information provided by the manufacturer (Thermo Co. USA), RNaseA can degrade dsRNA under low salt conditions as used in this study (**Supplementary Figure 1**). At the same time, we checked some references which have reported this biochemical finding as well. We also revised a little in the Methods section (**Line 28-30**).

thermo
scientific

PRODUCT INFORMATION

**RNase A,
DNase and Protease-free**

Pub. No. MAN0012003
Rev. Date 22 September 2017 (Rev. C.00)

#EN0531

Assembling Lot 00000000

Filling Lot 00000000

Expiry Date MM.YYYY

Store at -25 °C to -15 °C

Components	#EN0531
RNase A, DNase and Protease-free, 10 mg/mL	10 mg

www.thermofisher.com
For Research Use Only. Not for use in diagnostic procedures.

Description

RNase A is an endoribonuclease that specifically degrades single-stranded RNA at C and U residues. It cleaves the phosphodiester bond between the 5'-ribose of a nucleotide and the phosphate group attached to the 3'-ribose of an adjacent pyrimidine nucleotide. The resulting 2', 3'-cyclic phosphate is hydrolyzed to the corresponding 3'-nucleoside phosphate (1, 2).

Applications

- Plasmid and genomic DNA preparation (3, 4).
- Removal of RNA from recombinant protein preparations.
- Ribonuclease protection assays. Used in conjunction with RNase T1 (3).
- Mapping single-base mutations in DNA or RNA (5, 6).

Source

Bovine pancreas.

Molecular Weight

13.7 kDa monomer.

Concentration

Protein concentration is determined by measuring the absorbance at 278 nm using molar absorption coefficient $\epsilon=9800 \text{ M}^{-1}\text{cm}^{-1}$ (7).

Definition of Activity Unit

One unit of the enzyme causes an increase in absorbance of 1.0 at 260 nm when yeast RNA is hydrolyzed at 37 °C and pH 5.0. Fifty units are approximately equivalent to 1 Kunitz unit (8).

Specific activity

$\geq 5000 \text{ U/mg protein}$ ($\geq 100 \text{ Kunitz units/mg protein}$).

Storage Buffer

The enzyme is supplied in: 50 mM Tris-HCl (pH 7.4) and 50% (v/v) glycerol.

Inhibition and Inactivation

- Inhibitors: the most potent inhibitor is a mammalian ribonuclease inhibitor, e.g., Thermo Scientific RiboLock RNase Inhibitor (#EO0381).
- Other inhibitors: uridine 2',3'-cyclic vanadate, 5'-diphosphoadenosine 3'-phosphate and 5'-diphosphoadenosine 2'-phosphate (2), SDS, diethyl pyrocarbonate, 4 M guanidinium thiocyanate plus 0.1 M 2-mercaptoethanol and heavy metal ions.
- Not inactivated by heating, reliably removed by spin column or phenol/chloroform extraction.

Note

- Recommended concentration of RNase A is 1-100 $\mu\text{g/mL}$ depending on the application.
- The enzyme is active under a wide range of reaction conditions. At low salt concentrations (0 to 100 mM NaCl), RNase A cleaves single-stranded and double-stranded RNA as well the RNA strand in RNA-DNA hybrids. However, at NaCl concentrations of 0.3 M or higher, RNase A specifically cleaves single-stranded RNA (9).

Rev.10 
Reference:

[1] Nwokeoji, A.O. et al. Purification and characterisation of dsRNA using ion pair reverse phase chromatography and mass spectrometry. *J Chromatogr A*. **1484**:14-25 (2017).

● **Reviewer #3 (Remarks to the Author)**

Query 1. I am confused about the number of times and the number of locations in which the field experiment(s) was (were) conducted. This confusion results from reading the manuscript and in the response to my initial query. Please clarify. I strongly suspect there was one trial of the experiment. Please clarify.

We greatly appreciate your valuable feedback. The field trials were at different geographical locations (different fields), sowed on close dates during the April 1 to 10 period, and cellobiose/dsRNA/SPc were treated on same dates during the July 1-29 period, because the first signs of disease symptoms were occurred at same time. therefore, there were three trials of the experiments. The statistician analysis was truly independent. We detailed and revised this part in the Methods section (**Lines 168-172**).

● **Reviewer #4 (Remarks to the Author)**

The authors have done much to assuage this reviewer's concerns. A few remain:

Query 1.- The authors claim that the SPc/Cellobiose/dsRNA is significantly more effective than other treatments at controlling disease. Figure 3 a,c,d,e,f seem to indicate that dsRNA/SPc, Cellobiose/SPc, and Cellobiose/SPc/dsRNA all control disease about equally. N=3 replicates is quite small for a plant disease experiment so I am not convinced the difference between these three treatments is significant. I suggest change the wording to reflect that all three behave similarly.

Thanks for your comments. We changed the wording according to your suggestion (**Lines 146-147**). By the way, in the field experiment, Cellobiose/SPc/dsRNA complex showed excellent protective effect and had significant difference compared to the other treatments and control.

Query 2.- Do the authors observe any detrimental impacts to yield from their treatment? Immune

priming is no longer widely pursued commercially due to the trade-off between growth and immunity. If this is to be a commercially viable/relevant finding, this issue should be addressed.

In the greenhouse and field, we haven't observed any negative effect on the growth of potato plants; there were no growth-defect phenotypes. In the field, the preliminary yield measurement in the field showed that the application of the multicomponent nano-bioprotectant has the best increasing effects in potato yield compared with the other treatments and control. We have added some description in the manuscript (**Line 150**).

Reviewers' Comments:

Reviewer #2:

Remarks to the Author:

I have no more problems to see the manuscript published. Concerning the RNase A issue. After consulting " Methods in Enzymology "and some biochemical experts I agree that the oligomeric form (!) of RNase A can also degrade dsRNA but is not specific for it! The authors should check whether this is consistent with their wording.

Reviewer #3:

Remarks to the Author:

The authors have demonstrated a technique to mitigate disease caused by *Phytophthora infestans*. It looks intriguing.

The authors have satisfactorily addressed my questions.

William E. Fry

Reviewer #4:

Remarks to the Author:

The authors have adequately addressed my concerns about their work.

RESPONSE TO REVIEWERS' COMMENTS

Thanks to the Nature Communications' editors for granting us the opportunity to improve our manuscript. We would like to thank all reviewers for their valuable comments to improve our study. We also appreciate the timeframe to achieve our revision with the primary goal of reaching the high-quality standards expected in this journal. We have tried our best to conduct revisions, please see the line-by-line Response to Reviewers' comments for the details.

- **Reviewer #2 (Remarks to the Author)**

Query 1. I have no more problems to see the manuscript published. Concerning the RNase A issue. After consulting " Methods in Enzymology "and some biochemical experts I agree that the oligomeric form (!) of RNase A can also degrade dsRNA but is not specific for it! The authors should check whether this is consistent with their wording.

Thanks for your suggestion. We have added the specific condition description and changed the wording to "RNase A can also degrade dsRNA at low salt concentration" in the manuscript (see line 70).

- **Reviewer #3 (Remarks to the Author):**

Query 1. The authors have demonstrated a technique to mitigate disease caused by *Phytophthora infestans*. It looks intriguing. The authors have satisfactorily addressed my questions.

We greatly appreciate your valuable feedback.

- **Reviewer #4 (Remarks to the Author):**

Query 1. The authors have adequately addressed my concerns about their work.

Thanks for your comments.